# HYTREL: Hypergraph-enhanced Tabular Data Representation Learning

**Pei Chen**[1][*], **Soumajyoti Sarkar**[2], **Leonard Lausen**[2],
**Balasubramaniam Srinivasan**[2], **Sheng Zha**[2], **Ruihong Huang**[1], **George Karypis**[2]
[1]Texas A&M University, [2]Amazon Web Services
{chenpei,huangrh}@tamu.edu,
{soumajs,lausen,srbalasu,zhasheng,gkarypis}@amazon.com

## Abstract

Language models pretrained on large collections of tabular data have demonstrated their effectiveness in several downstream tasks. However, many of these models do not take into account the row/column permutation invariances, hierarchical structure, etc. that exist in tabular data. To alleviate these limitations, we propose **HYTREL**, a tabular language model, that captures the permutation invariances and three more ***structural properties*** of tabular data by using hypergraphs–where the table cells make up the nodes and the cells occurring jointly together in each row, column, and the entire table are used to form three different types of hyperedges. We show that HYTREL is maximally invariant under certain conditions for tabular data, i.e., two tables obtain the same representations via HYTREL *iff* the two tables are identical up to permutations. Our empirical results demonstrate that HYTREL **consistently** outperforms other competitive baselines on four downstream tasks with minimal pretraining, illustrating the advantages of incorporating the inductive biases associated with tabular data into the representations. Finally, our qualitative analyses showcase that HYTREL can assimilate the table structures to generate robust representations for the cells, rows, columns, and the entire table. [1]

## 1 Introduction

Tabular data that is organized in bi-dimensional matrices are widespread in webpages, documents, and databases. Understanding tables can benefit many tasks such as table type classification, table similarity matching, and knowledge extraction from tables (e.g., column annotations) among others. Inspired by the success of pretrained language models in natural language tasks, recent studies [Yin et al., 2020, Yang et al., 2022] proposed Tabular Language Models (TaLMs) that perform pretraining on tables via self-supervision to generate expressive representations of tables for downstream tasks.

Among the TaLMs, many works [Herzig et al., 2020, Yin et al., 2020, Deng et al., 2020, Iida et al., 2021] serialize tables to a sequence of tokens for leveraging existing pretrained language model checkpoints and textual self-supervised objectives like the Masked Language Modeling. However, due to the linearization of tables to strings, these models do not explicitly incorporate the structural properties of a table, e.g., the invariances to arbitrary permutations of rows and columns (independently). Our work focuses on obtaining representations of tables that take table structures into account. We hypothesize that incorporating such properties into the table representations will benefit many downstream table understanding tasks.

---

[*]Work done as an intern at Amazon Web Services.
[1]Code is available at: https://github.com/awslabs/hypergraph-tabular-lm

37th Conference on Neural Information Processing Systems (NeurIPS 2023).

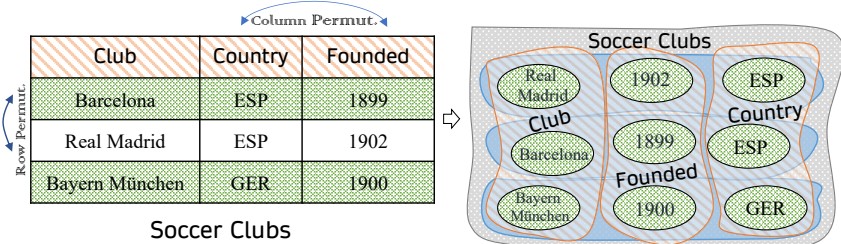

Figure 1: An example of modeling a table as a hypergraph. Cells make up the nodes and the cells in each row, column, and the entire table form hyperedges. The table caption and the header names are used for the names of the table and column hyperedges. The hypergraph keeps the four structural properties of tables, e.g., the invariance property of the table as the row/column permutations result in the same hypergraph.

**Motivation:** Tabular data is structurally different in comparison to other data modalities such as images, audio, and plain texts. We summarize four *structural properties* present in the tables below:

- Most tables are invariant to row/column permutations. This means, in general, if we arbitrarily (and independently) permute the rows or columns of a table, it is still an equivalent table. For other tables with an explicit ordering of rows or columns, we can make them permutation invariant by appropriately adding a ranking index as a new column or row.
- Data from a single column are semantically similar – for example, they oftentimes have the same semantic types. Similarly, the cells within a single row together describe the attributes of a sample within the table and the cells cannot be treated in silos.
- The interactions within cells/rows/columns are not necessarily pairwise, i.e., the cells within the same row/column, and rows/columns from the same table can have high-order multilateral relations [**?**].
- Information in tables is generally organized in a hierarchical fashion where the information at the table-level can be aggregated from the column/row-level, and further from the cell-level.

However, the linearization-based approaches are not designed to explicitly capture most of the above properties. We aim to address the limitation by modeling all the aforementioned structural properties as inductive biases while learning the table representations.

**Our Approach:** In line with recent studies [Deng et al., 2020, Yang et al., 2022, Wang et al., 2021a] which have elucidated upon the importance of the structure of a table, we propose the HYTREL that uses hypergraphs to model the tabular data. We propose a modeling paradigm that aims *capture all of the four properties* directly. Figure 1 provides an example of how a hypergraph is constructed from a table. As observed, converting a table into a hypergraph allows us to incorporate the first two properties inherent to the nature of hypergraphs. Hypergraphs seamlessly allow the model to incorporate row/column permutation invariances, as well as interactions among the cells within the same column or row. Moreover, the proposed hypergraph structure can capture the high-order (not just pairwise) interactions for the cells in a column or a row, as well as from the whole table, and an aggregation of hyperedges can also help preserve the hierarchical structure of a table.

**Contributions:** Our theoretical analysis and empirical results demonstrate the advantages of modeling the four structural properties. We first show that HYTREL is maximally invariant when modeling tabular data (under certain conditions), i.e. if two tables get the same representations via the hypergraph table learning function, then the tables differ only by row/column permutation (independently) actions and vice versa. Empirically, we pretrain HYTREL on publicly available tables using two self-supervised objectives: a table content based ELECTRA[2] objective [Clark et al., 2020, Iida et al., 2021] and a table structure dependent contrastive objective [Wei et al., 2022]. The evaluation of the pretrained HYTREL model on four downstream tasks (two knowledge extraction tasks, a table type detection task, and a table similarity prediction task) shows that HYTREL can achieve state-of-the-art performance.

We also provide an extensive qualitative analysis of HYTREL–including visualizations that showcase that (a) HYTREL representations are robust to arbitrary permutations of rows and columns (independently), (b) HYTREL can incorporate the hierarchical table structure into the representations, (c)

HYTREL can achieve close to state-of-the-art performance even without pretraining, and the model is extremely efficient with respect to the number epochs for pretraining in comparison to prior works, further demonstrating the advantages of HYTREL in modeling the structural properties of tabular data. In Appendix B, we provide additional analysis that demonstrates HYTREL's ability to handle input tables of arbitrary size and underscore the importance of the independent row/column permutations.

## 2   HYTREL Model

Formally, a table in our work is represented as $\mathcal{T} = [M, H, R]$, where $M$ is the caption, $H = [h_1, h_2, h_3, ..., h_m]$ are the $m$ column headers, $R$ represents the $n$ rows $[R_1, R_2, R_3, ..., R_n]$. Each row $R_i$ has $m$ cells $[c_{i1}, c_{i2}, c_{i3}, ..., c_{im}]$. The caption, header, and cell values can be regarded as sentences that contain several words. We note that each cell $c_{ij}$ or header $h_j$ also belongs to the corresponding column $C_j$. We use $C = [C_1, C_2, C_3, ..., C_m]$ to represent all the columns that include the headers, so a table can also be defined as $\mathcal{T} = [M, C]$.

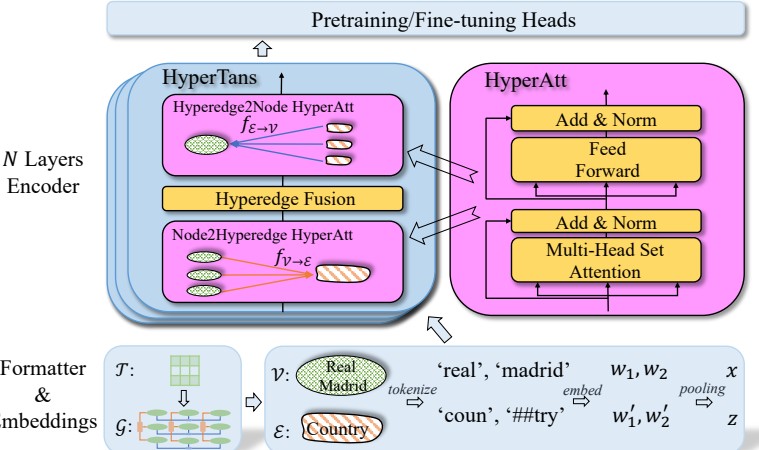

Figure 2: The overall framework of HYTREL. We first turn a table $\mathcal{T}$ into a hypergraph $\mathcal{G}$ and then initialize the embeddings of the nodes $\mathcal{V}$ and hyperedges $\mathcal{E}$. After that, we encode the hypergraph using stacked multiple-layer hypergraph-structure-aware transformers (HyperTrans). Each HyperTrans layer has two attention blocks that work on hypergraph (HyperAtt) and one Hyperedge Fusion block. Lastly, we use the node and hyperedge representations from the final layer for pretraining and fine-tuning.

### 2.1   Formatter & Embedding Layer

The formatter transforms a table into a hypergraph. As shown in Figure 1, given a table $\mathcal{T}$, we construct a corresponding hypergraph $\mathcal{G} = (\mathcal{V}, \mathcal{E})$, where $\mathcal{V}, \mathcal{E}$ denote the set of nodes and hyperedges respectively. We treat each cell $c_{ij}$ as a node $v_{ij} \in \mathcal{V}$, and each row $R_i$, each column $C_j$, and the entire table $\mathcal{T}$ as hyperedges $e_i^c, e_j^r, e^t \in \mathcal{E}$ $(1 \leq i \leq n, 1 \leq j \leq m)$, respectively. As a part of our hypergraph construction, each cell node $v_{ij}$ is connected to 3 hyperedges: its column hyperedge $e_i^c$, row hyperedge $e_j^r$, and the table hyperedge $e^t$. The hypergraph can be conveniently be represented as an incidence matrix $\mathbf{B} \in \{0, 1\}^{mn \times (m+n+1)}$, where $\mathbf{B}_{ij} = 1$ when node $i$ belong to hyperedge $j$ and $\mathbf{B}_{ij} = 0$ otherwise.

An embedding layer is then employed over the nodes and hyperedges . Each node $v_{ij}$ corresponds to a cell $c_{ij}$ that has several tokens, and we obtain the feature vector $\mathbf{X}_{v_{ij}, :} \in \mathbb{R}^F$ for a given node by feeding its constituent cell tokens into the embedding layer and then averaging the embeddings over the tokens. After obtaining node embeddings $\mathbf{X} \in \mathbb{R}^{nm \times F}$ for all nodes, a similar transformation is applied over hyperedges. For different hyperedge types, we use different table content for their initialization : for a column hyperedge $e_i^c$, we use all the tokens from the corresponding header $h_i$. For the table hyperedge $e^t$, we use the the entire caption associated with $M$. For the row hyperedge $e_j^r$,

---

[2]We use the name ELECTRA following Iida et al. [2021] but it is different from the ELECTRA objective used in language modeling [Clark et al., 2020]. We do not utilize an auxiliary model to generate corruption.

when no semantic information is available, we randomly initialize them as $\mathbf{S}_{e_j^r,:} \in \mathbb{R}^F$. Performing the above operations yields an initialization for all the hyperedge embeddings $\mathbf{S} \in \mathbb{R}^{(m+n+1) \times F}$.

## 2.2 Hypergraph Encoder

After embedding, we propose to use a structure-aware transformer module (`HyperTrans`) to encode the hypergraphs. `HyperTrans` encoder can encode the table content, structure, and relations among the table elements (including cells, headers, captions, etc.). As shown in Figure 2, one layer of the `HyperTrans` module is composed of two hypergraph attention blocks (`HyperAtt`, $f$) [Chien et al., 2022] that interact with the node and hyperedge representations, and one `Hyperedge Fusion` block. The first `HyperAtt` block is the `Node2Hyperedge` attention block as defined below:

$$\tilde{\mathbf{S}}_{e,:}^{(t+1)} = f_{\mathcal{V} \to \mathcal{E}} \left( K_{e,\mathbf{X}^{(t)}} \right) \tag{1}$$

Where $f_{\mathcal{V} \to \mathcal{E}}$ is a hypergraph attention function defined from nodes to hyperedges. $K_{e,\mathbf{X}} = \{\mathbf{X}_{v,:} : v \in e\}$ denotes the sets of hidden node representations included in the hyperedge $e$. The `Node2Hyperedge` block will aggregate information to hyperedge $e$ from its constituent nodes $v \in e$.

We then use a `Hyperedge Fusion` module (a Multilayer Perceptron Network, MLP) to propagate the hyperedge information from the last step, as defined below:

$$\mathbf{S}_{e,:}^{(t+1)} = \text{MLP} \left( \mathbf{S}_{e,:}^{(t)}; \tilde{\mathbf{S}}_{e,:}^{(t+1)} \right) \tag{2}$$

A second `HyperAtt` block `Hyperedge2Node` then aggregates information from a hyperedge to its constituent nodes as follows:

$$\mathbf{X}_{v,:}^{(t+1)} = f_{\mathcal{E} \to \mathcal{V}} \left( L_{v,\mathbf{S}^{(t+1)}} \right) \tag{3}$$

Where $f_{\mathcal{E} \to \mathcal{V}}$ is another hypergraph attention function defined from hyperedges to nodes. $L_{v,\mathbf{S}} = \{\mathbf{S}_{e,:} : v \in e\}$ is defined as the sets of hidden representations of hyperedges that contain the node $v$.

As for the `HyperAtt` block $f$, similar to transformer [Vaswani et al., 2017], it is composed of one multi-head attention, one Position-wise Feed-Forward Network (FFN), two-layer normalization (LN) [Ba et al., 2016] and two skip connections [He et al., 2016], as in Figure 2. However, we do not use the self-attention [Vaswani et al., 2017] mechanism from the transformer model because it is not designed to keep the invariance structure of tables or hypergraphs. Inspired by the deep set models [Zaheer et al., 2017, Lee et al., 2019], we use a set attention mechanism that can keep the permutation invariance of a table. We define `HyperAtt` $f$ as follows:

$$f_{\mathcal{V} \to \mathcal{E} \text{ or } \mathcal{E} \to \mathcal{V}}(\mathbf{I}) := \text{LN}(\mathbf{Y} + \text{FFN}(\mathbf{Y})) \tag{4}$$

Where $\mathbf{I}$ is the input node or hyperedge representations. The intermediate representations $\mathbf{Y}$ is obtained by:

$$\mathbf{Y} = \text{LN}\left( \omega + \text{SetMH}(\omega, \mathbf{I}, \mathbf{I}) \right) \tag{5}$$

Where $\text{SetMH}$ is the multi-head set attention mechanism defined as:
$$\text{SetMH}(\omega, \mathbf{I}, \mathbf{I}) = \|_{i=1}^{h} \mathbf{O}_i \tag{6}$$

and

$$\mathbf{O}_i = \text{Softmax}\left( \omega_i \left( \mathbf{I} \mathbf{W}_i^K \right)^T \right) \left( \mathbf{I} \mathbf{W}_i^V \right) \tag{7}$$

Where $\omega$ is a learnable weight vector as the query and $\omega := \|_{i=1}^{h} \omega_i$, $\mathbf{W}_i^K$ and $\mathbf{W}_i^V$ are the weights for the key and value projections, $\|$ means concatenation.

So the `HyperTrans` module will update node and hyperedge representations alternatively. This mechanism enforces the table cells to interact with the columns, rows, and the table itself. Similar to $\text{BERT}_{base}$ [Devlin et al., 2019] and $\text{TaBERT}_{base}$ [Yin et al., 2020] , we stack 12 layers of `HyperTrans`.

## 2.3 Invariances of the HYTREL Model

Let $\phi : \mathcal{T} \mapsto \mathbf{z} \in \mathbb{R}^d$ be our target function which captures the desired row/column permutation invariances of tables (say for tables of size $n \times m$). Rather than working on the table $\mathcal{T}$ directly, the proposed HYTREL model works on a hypergraph (via Eqns (1-5)) that has an incidence matrix $\mathbf{B}$ of size $mn \times (m + n + 1)$. Correspondingly, we shall refer to HYTREL as a function $g : \mathbf{B} \mapsto \mathbf{y} \in \mathbb{R}^k$.

In this section we will make the connections between the properties of the two functions $\phi$ and $g$, demonstrating a maximal invariance between the two–as a result of which we prove that our HYTREL can also preserve the permutation invariances of the tables. First, we list our assumptions and resultant properties of tabular data. Subsequently, we present the maximal invariance property of $\phi$ and our hypergraph-based learning framework $g$. As a part of our notation, we use $[n]$ to denote $\{1, 2, \ldots, n\}$. Preliminaries and all detailed proofs are presented in the Appendix A.1 and A.2 respectively.

**Assumption 2.1.** For any table $(\mathcal{T}_{ij})_{i \in [n], j \in [m]}$ (where $i, j$ are indexes of the rows, columns), an arbitrary group action $a \in \mathbb{S}_n \times \mathbb{S}_m$ acting appropriately on the rows and columns leaves the target random variables associated with tasks on the entire table unchanged.

This assumption is valid in most real-world tables – as reordering the columns and the rows in the table oftentimes doesn't alter the properties associated with the entire table (e.g. name of the table, etc). As noted earlier, for tables with an explicit ordering of rows or columns, we can make them permutation invariant by adding a ranking index as a new column or row appropriately. To model this assumption, we state a property required for functions acting on tables next.

*Property* 1. A function $\phi : \mathcal{T} \mapsto \mathbf{z} \in R^d$ which satisfies Assumption 2.1 and defined over tabular data must be invariant to actions from the (direct) product group $\mathbb{S}_n \times \mathbb{S}_m$ acting appropriately on the table i.e. $\phi(a \cdot \mathcal{T}) = \phi(\mathcal{T}) \ \forall a \in \mathbb{S}_n \times \mathbb{S}_m$.

However, HYTREL (or the function $g$ via hypergraph modeling) through Eqns (1-5)) models invariances of the associated incidence matrix to the product group $\mathbb{S}_{mn} \times \mathbb{S}_{m+n+1}$ (proof presented in the appendix). To make the connection between the two, we present the maximal invariance property of our proposed HYTREL model.

**Theorem 2.2.** *A continuous function $\phi : \mathcal{T} \mapsto \mathbf{z} \in \mathbb{R}^d$ over tables is maximally invariant when modeled as a function $g : \mathbf{B} \mapsto \mathbf{y} \in \mathbb{R}^k$ over the incidence matrix of a hypergraph $\mathcal{G}$ constructed per Section 2.1 (Where $g$ is defined via Eqns (1-5)) if $\exists$ a bijective map between the space of tables and incidence matrices (defined over appropriate sizes of tables, incidence matrices). That is, $\phi(\mathcal{T}_1) = \phi(\mathcal{T}_2)$ iff $\mathcal{T}_2$ is some combination of row and/or column permutation of $\mathcal{T}_1$ and $g(\mathbf{B_1}) = g(\mathbf{B_2})$ where $\mathbf{B}_1, \mathbf{B}_2$ are the corresponding (hypergraph) incidence matrices of tables $\mathcal{T}_1, \mathcal{T}_2$.*

*Proof Sketch*: Detailed proof is provided in Appendix A.2. The above theorem uses Lemma 1 from [Tyshkevich and Zverovich, 1996] and applies the Weisfeiler-Lehman test of isomorphism over the star expansion graphs of the hypergraphs toward proving the same.

As a consequence of Theorem 2.2, two tables identical to permutations will obtain the same representation, which has been shown to improve generalization performance [Lyle et al., 2020].

## 2.4 Pretraining Heads

**ELECTRA Head**: In the ELECTRA pretraining setting, we first corrupt a part of the cells and the headers from a table and then predict whether a given cell or header has been corrupted or not Iida et al. [2021]. Cross-entropy loss is used together with the binary classification head.

**Contrastive Head**: In the contrastive pretraining setting, we randomly corrupt a table-transformed hypergraph by masking a portion of the connections between nodes and hyperedges, as inspired by the hypergraph contrastive learning [Wei et al., 2022]. For each hypergraph, we corrupt two augmented views and use them as the positive pair, and use the remaining in-batch pairs as negative pairs. Following this, we contrast the table and column representations from the corresponding hyperedges. The InfoNCE [van den Oord et al., 2018] objective is used for optimization as in 8.

$$loss = -\log \frac{\exp\left(\boldsymbol{q} \cdot \boldsymbol{k}_+/\tau\right)}{\sum_{i=0}^{K} \exp\left(\boldsymbol{q} \cdot \boldsymbol{k}_i/\tau\right)} \tag{8}$$

where $(\boldsymbol{q}, \boldsymbol{k}_+)$ is the positive pair, and $\tau$ is a temperature hyperparameter.

# 3 Experiments

## 3.1 Pre-training

**Data** In line with previous TaLMs [Yin et al., 2020, Iida et al., 2021], we use tables from Wikipedia and Common Crawl for pretraining. We utilize preprocessing tools provided by Yin et al. [2020] and collect a total of 27 million tables (1% are sampled and used for validation).[3] During pretraining, we truncate large tables and retain a maximum of 30 rows and 20 columns for each table, with a maximum of 64 tokens for captions, column names, and cell values. It is important to note that the truncation is solely for efficiency purposes and it does not affect HYTREL's ability to deal with large tables, as elaborated in appendix B.1.

**Settings** With the ELECTRA pretraining objective, we randomly replace 15% of the cells or headers of an input table with values that are sampled from all the pretraining tables based on their frequency, as recommended by Iida et al. [2021]. With the contrastive pretraining objective, we corrupted 30% of the connections between nodes and hyperedges for each table to create one augmented view. The temperature $\tau$ is set as 0.007. For both objectives, we pretrain the HYTREL models for 5 epochs. More details can be found the Appendix C.1.

## 3.2 Fine-tuning[4]

After pretraining, we use the HYTREL model as a table encoder to fine-tune downstream table-related tasks. In order to demonstrate that our model does not heavily rely on pretraining or on previous pretrained language models, we also fine-tune the randomly initialized HYTREL model for comparison. In this section, we introduce the evaluation tasks and the datasets. We choose the following four tasks that rely solely on the table representations since we want to test the task-agnostic representation power of our model and avoid training separate encoders for texts (e.g., questions in table QA tasks) or decoders for generations. As mentioned, our encoder can be used in all these scenarios and we leave its evaluation in other table-related tasks as future work.

**Column Type Annotation** (CTA) task aims to annotate the semantic types of a column and is an important task in table understanding which can help many knowledge discovery tasks such as entity recognition and entity linking. We use the column representations from the final layer of HYTREL with their corresponding hyperedge representations for making predictions. We evaluate HYTREL on the TURL-CTA dataset constructed by Deng et al. [2020].

**Column Property Annotation** (CPA) task aims to map column pairs from a table to relations in knowledge graphs. It is an important task aimed at extracting structured knowledge from tables. We use the dataset TURL-CPA constructed by Deng et al. [2020] for evaluation.

**Table Type Detection** (TTD) task aims to annotate the semantic type of a table based on its content. We construct a dataset using a subset from the public WDC Schema.org Table Corpus.

**Table Similarity Prediction** (TSP) task aims at predicting the semantic similarity between tables and then classifying a table pair as similar or dissimilar. We use the PMC dataset proposed by Habibi et al. [2020] for evaluation.

## 3.3 Baselines

**TaBERT** [Yin et al., 2020] is a representative TaLM that flattens the tables into sequences and jointly learns representations for sentences and tables by pretraining the model from the BERT checkpoints. *K=1* and *K=3* are the two variants based on the number of rows used.

**TURL** [Deng et al., 2020] is another representative TaLM that also flattens the tables into sequences and pretrains from TinyBERT [Jiao et al., 2020] checkpoints. It introduces a vision matrix to incorporate table structure into the representations.

**Doduo** [Suhara et al., 2022] is a state-of-the-art column annotation system that fine-tunes the BERT and uses table serialization to incorporate table content.

---

[3] As the version of Wikipedia used by [Yin et al., 2020] is not available now, we use an updated version so we collect slightly more tables than previous TaLMs.

[4] More details about experimental settings, the datasets, and the baselines can be found the Appendix C.2

| Systems | Column Type Annotation | Column Property Annotation |
|---|---|---|
| Sherlock | 88.40 / 70.55 / 78.47 | - |
| BERT$_{base}$ | - | 91.18 / 90.69 / 90.94 |
| TURL + metadata | 92.75 / 92.63 / 92.69 | 92.90 / 93.80 / 93.35 |
| Doduo + metadata | 93.25 / 92.34 / 92.79 | 91.20 / 94.50 / 92.82 |
| TaBERT$_{base}$(K=1) | $91.40_{\pm0.06}$ / $89.49_{\pm0.21}$ / $90.43_{\pm0.11}$ | $92.31_{\pm0.24}$ / $90.42_{\pm0.53}$ / $91.36_{\pm0.30}$ |
| *w/o* Pretrain | $90.00_{\pm0.14}$ / $85.50_{\pm0.09}$ / $87.70_{\pm0.10}$ | $89.74_{\pm0.40}$ / $68.74_{\pm0.93}$ / $77.84_{\pm0.64}$ |
| TaBERT$_{base}$(K=3) | $91.63_{\pm0.21}$ / $91.12_{\pm0.25}$ / $91.37_{\pm0.08}$ | $92.49_{\pm0.18}$ / $92.49_{\pm0.22}$ / $92.49_{\pm0.10}$ |
| *w/o* Pretrain | $90.77_{\pm0.11}$ / $87.23_{\pm0.22}$ / $88.97_{\pm0.12}$ | $90.10_{\pm0.17}$ / $84.83_{\pm0.89}$ / $87.38_{\pm0.48}$ |
| HYTREL *w/o* Pretrain | $\mathbf{92.92}_{\pm0.11}$ / $92.50_{\pm0.10}$ / $92.71_{\pm0.08}$ | $92.85_{\pm0.35}$ / $91.50_{\pm0.54}$ / $92.17_{\pm0.38}$ |
| HYTREL *w/* ELECTRA | $92.85_{\pm0.21}$ / $\mathbf{94.21}_{\pm0.09}$ / $\mathbf{93.53}_{\pm0.10}$ | $92.88_{\pm0.24}$ / $\mathbf{94.07}_{\pm0.27}$ / $\mathbf{93.48}_{\pm0.12}$ |
| HYTREL *w/* Contrastive | $92.71_{\pm0.20}$ / $93.24_{\pm0.08}$ / $92.97_{\pm0.13}$ | $\mathbf{93.01}_{\pm0.40}$ / $93.16_{\pm0.40}$ / $93.09_{\pm0.17}$ |

Table 1: Test results on the CTA and CPA tasks (Precision/Recall/F1 Scores,%). The results of TaBERT and HYTREL are from the average of 5 system runs with different random seeds. For fair comparisons, we use the results of TURL and Doduo with metadata, i.e., captions and headers.

| Systems | Table Type Detection | Table Similarity Prediction | |
|---|---|---|---|
| | Accuracy | Precision/Recall/F1 | Accuracy |
| TFIDF+Glove+MLP | - | 87.36 / 83.81 / 84.47 | 85.06 |
| TabSim | - | 88.65 / 85.45 / 86.13 | 87.05 |
| TaBERT$_{base}$(K=1) | $93.11_{\pm0.31}$ | $87.04_{\pm0.64}$ / $85.34_{\pm0.93}$ / $86.18_{\pm1.13}$ | $87.35_{\pm1.42}$ |
| *w/o* Pretrain | $85.04_{\pm0.41}$ | $33.61_{\pm12.70}$ / $50.31_{\pm12.75}$ / $40.30_{\pm12.03}$ | $63.45_{\pm10.11}$ |
| TaBERT$_{base}$(K=3) | $95.15_{\pm0.14}$ | $87.76_{\pm0.64}$ / $86.97_{\pm0.59}$ / $87.36_{\pm0.95}$ | $88.29_{\pm0.98}$ |
| *w/o* Pretrain | $89.88_{\pm0.26}$ | $82.96_{\pm1.84}$ / $81.16_{\pm1.45}$ / $82.05_{\pm1.02}$ | $82.57_{\pm1.20}$ |
| HYTREL *w/o* Pretrain | $93.84_{\pm0.17}$ | $88.94_{\pm1.83}$ / $85.72_{\pm1.52}$ / $87.30_{\pm1.02}$ | $88.38_{\pm1.43}$ |
| HYTREL *w/* ELECTRA | $\mathbf{95.81}_{\pm0.19}$ | $87.35_{\pm0.42}$ / $87.29_{\pm0.84}$ / $87.32_{\pm0.50}$ | $88.29_{\pm0.49}$ |
| HYTREL *w/* Contrastive | $94.52_{\pm0.30}$ | $\mathbf{89.41}_{\pm0.58}$ / $\mathbf{89.10}_{\pm0.90}$ / $\mathbf{89.26}_{\pm0.53}$ | $\mathbf{90.12}_{\pm0.49}$ |

Table 2: Test results on the TTD (Accuracy Score,%) and TSP (Precision/Recall/F1 Scores,%) tasks. The results of TaBERT and HYTREL are from the average of 5 system runs with different random seeds.

## 3.4 Main Results

The results are presented in Tables 1 and 2. Overall, HYTREL consistently outperforms the baselines and achieve superior performance. A salient observation is that our model (even without pretraining) can achieve close to state-of-the-art performance. In comparison, we notice that the performance slumps significantly for TaBERT without pretraining. This phenomenon empirically demonstrates the advantages of modeling the table structures as hypergraphs over the other methods that we compare.

Additionally, we observe that the two pretraining objectives help different tasks in different ways. For the CTA, CPA, and TTD tasks, the two objectives can help HYTREL further improve its performance. In general, the ELECTRA objective performs better than the contrastive objective. These results are also in line with the representation analysis in Section 4.2 where we observe that the ELECTRA objective tends to learn table structure better than the contrastive objective. However, for the TSP task, we observe that the contrastive objective can help the HYTREL model while the ELECTRA objective fails to bring any improvement. One possible reason for the ineffectiveness of the ELECTRA objective could be its inability to transfer well across domains. HYTREL pretrained with tables from Wikipedia and Common Crawl could not transfer well to the medical domain PMC dataset. As for the improvement observed from the contrastive objective, the reason could be that contrastive learning that uses similarity metrics in the objective function can naturally help the similarity prediction task.

**Scalability:** As stated in Section 3, we have truncated large tables during pretraining. However, this truncation does not hinder the ability of HYTREL to handle large table inputs in downstream tasks. In Appendix B.1, we present additional experiments demonstrating that: (a) HYTREL can effectively process tables of any size as inputs, and (b) down-sampling can be a favorable strategy when working with large input tables, significantly improving efficiency without compromising performance.

# 4 Qualitative Analysis

## 4.1 HYTREL Learns Permutation Robust Representations

We sample 5,000 tables from the validation data for analysis. We analyze the impact of applying different permutations to a table, including permuting rows, columns, and both rows/columns independently.

Toward our analysis, we measure the Euclidean distance (L2 Norm) of the representations (cells, rows, columns and tables). As shown in Figure 3, the distance is almost always 0 for the HYTREL model because of its explicit invariance-preserving property. On the other hand, for the TaBERT model, the distance is not trivial. We observe that when more rows (K=3) are enabled, the value of the L2 norm increases as we introduce different permutations. Moreover, we also observe that permuting the columns has a greater impact on the L2 norm than the row permutations. A combination of rows and columns permutations has the largest impact among all three actions. Note that when K=1 with just one row, the effect of row permutation is disabled for TaBERT.

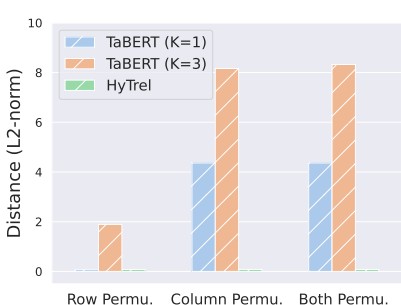

Figure 3: Average distance between table element representations before and after permutations. The HYTREL is immune to the permutations while the TaBERT is sensitive to them.

## 4.2 HYTREL Learns the Underlying Hierarchical Table Structure

Next, we demonstrate that the HYTREL model has learned the hierarchical table structure into its representations, as we target at. We use t-SNE [Van der Maaten and Hinton, 2008] for the visualization of different table elements from the same 5,000 validation tables, as shown in Figure 4.

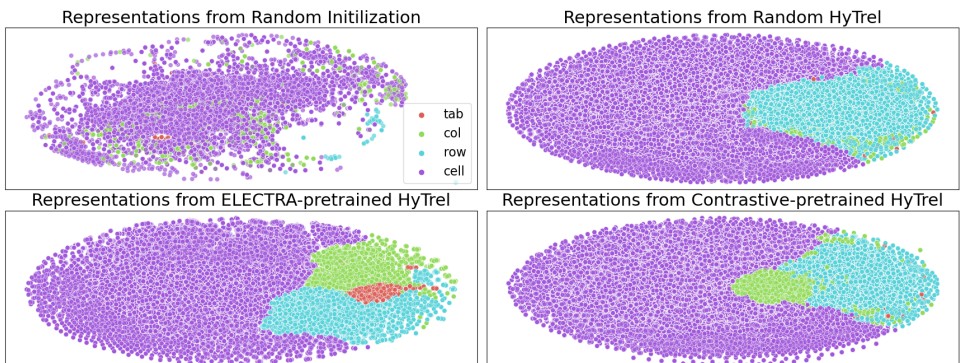

Figure 4: t-SNE visualization of the representations learned by HYTREL. 'tab', 'col', 'row', and 'cell' are the representations for different table elements: tables, columns, rows, and cells.

We observe that with random initializations, different table elements cannot be distinguished properly. After the encoding of the randomly initialized HYTREL model, we start to observe noticeable differences for different table elements in the visualization space. Notably, the individual cell representations start to concentrate together and can be distinguished from high-level table elements (tables, columns, and rows) which occupy their separate places in the space of representations. We also notice that, by pretraining the HYTREL with the ELECTRA objective, all table elements can be well separated, showing that it incorporates the hierarchical table structure into its latent representations. As for the contrastive pretraining, we see that it can distinguish columns from rows as compared with randomly initialized HYTREL, but could not to well separate the table representations in comparison with the ELECTRA pretraining. This also partially explains the better performance of the ELECTRA pretraining in the CTA, CPA and TTD tasks in contrast to the contrastive pretraining.

## 4.3   HYTREL Demonstrates Effective Pretraining by Capturing the Table Structures

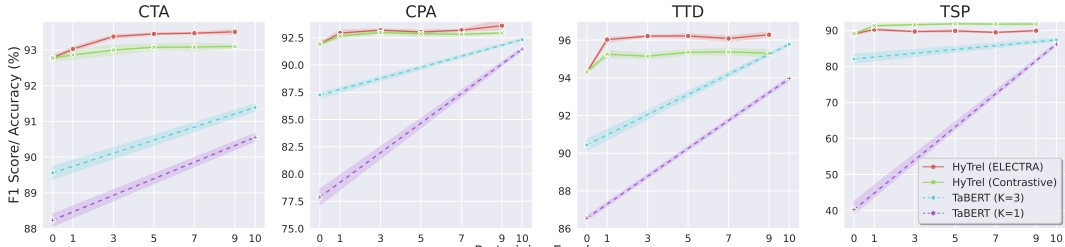

Figure 5: Performance with different pretraining checkpoints on the validation set of the four tasks. For the TaBERT models, we can only access the randomly initialized and fully pretrained (10 epochs) checkpoints. All results are from 5 system runs with different random seeds.

Our evaluation shows that the HYTREL model can perform well without pretraining, which demonstrates its training efficiency by modeling table structures. Here we further analyze how much pretraining is required for HYTREL to further enhance its performance, as compared with the baseline model. We plot the validation performance of the tasks evaluated at different pretraining checkpoints in Figure 5.

Overall, we can observe that the performance of HYTREL improves drastically during the first several pretraining epochs, and then saturates at about 5 epochs. With the minimal pretraining, HYTREL can outperform the fully pretrained TaBERT models. This demonstrates that our model does not require extensive pretraining to further improve its performance in contrast with previous TaLMs (e.g., TURL for 100 epochs, TaBERT for 10 epochs). Besides, we also observe from the curves that the ELECTRA objective consistently outperforms the contrastive objective for the CTA, CPA, and TTD tasks, but under-performs on the TSP task. Also, the ELECTRA objective has a negative impact on the TSP task when pretrained for longer duration, which is in line with our previous findings.

## 5   Related Work

There are two avenues of research that have studied in tabular data representation learning. The first group of studies focus on predicting labels (essentially one row) for classification and regression problems, using row information and column schema as input[Huang et al., 2020, Arik and Pfister, 2021, Somepalli et al., 2021, Gorishniy et al., 2021, Grinsztajn et al., 2022, Wang and Sun, 2022, Du et al., 2022, Wydmański et al., 2023, Liu et al., 2023]. These studies use gradient descent-based end-to-end learning and aim to outperform tree-based models through task-specific model pretraining and fine-tuning.

The second group of studies proposes TaLMs to retrieve task-agnostic tabular data representations for different downstream table understanding tasks. Drawing inspiration from textual Language Models like BERT [Devlin et al., 2019], many works [Herzig et al., 2020, Yin et al., 2020, Deng et al., 2020, Iida et al., 2021, Eisenschlos et al., 2021] serialize tables to a sequence of tokens, leveraging existing checkpoints and textual self-supervised objectives. However, the representations of the tables can not only be learned from table content and by utilizing table structures, similar to other forms of semi-structured data like code and HTML. Some contemporary works have noticed the importance of the table structures and introduce many techniques to learn a certain aspect of them, such as masking [Deng et al., 2020], coordinates [Wang et al., 2021a, Dash et al., 2022], and attention bias [Yang et al., 2022]. Our work belongs to this second group of studies and we propose to use hypergraphs to comprehensively model the rich table structures, and this is close to previous graph-based neural networks [Mueller et al., 2019, Wang et al., 2021b,c] where tables have been structured as graphs to incorporate row and column order information.

Table representation learning that focuses on joint text and table understanding is a separate field of research that partially overlaps with our work. Among them, some work [Herzig et al., 2020, Shi et al., 2022, Herzig et al., 2021, Glass et al., 2021, Yang et al., 2022] specialize in question-answering (QA) on tables and they jointly train a model that takes the question and the table structure as input together, allowing the pretraining to attend to the interactions of both texts and tables and boosting

the table-based QA tasks. Another branch of joint text and table understanding work focuses on text generation from tables[Parikh et al., 2020, Yoran et al., 2021, Wang et al., 2022, Andrejczuk et al., 2022], relying on an encoder-decoder model like T5 [Raffel et al., 2020] that can encode tables and decode free-form texts. In contrast to these studies, our work centers on the importance of structures in tables for table representation only, without extra text encoding or decoding.

Learning on hypergraphs has gone through a series of evolution [Agarwal et al., 2006, Yadati et al., 2019, Arya et al., 2020] in the way the hypergraph structure is modeled using neural networks layers. However, many of them collapse the hypergraph into a fully connected graph by clique expansion and cannot preserve the high-order interaction among the nodes. The recent development of permutation invariant networks [Zaheer et al., 2017, Lee et al., 2019] has enabled high-order interactions on the hypergraphs [Chien et al., 2022] that uses parameterized multi-set functions to model dual relations from node to hyperedges and vice versa. Closely related to the latest advancement, our HYTREL model adopts a similar neural message passing on hypergraphs to preserve the invariance property and high-order interactions of tables.

## 6 Limitations

The proposed HYTREL is a table encoder, and by itself cannot handle joint text and table understanding tasks like table QA and table-to-text generation. While it's possible to add text encoders or decoders for these tasks, it can potentially introduce additional factors that may complicate evaluating our hypothesis about the usefulness of modeling structural table properties. Moreover, the current model structure is designed for tables with simple column structures, like prior TaLMs, and cannot handle tables with complex hierarchical column structures. Additionally, HYTREL does not consider cross-table relations. Although we believe the hypergraph can generalize to model complicated column structures and cross-table interactions, we leave these aspects for future research.

## 7 Conclusion

In this work, we propose a tabular language model HYTREL that models tables as hypergraphs. It can incorporate the permutation invariances and table structures into the table representations. The evaluation on four table-related tasks demonstrates the advantages of learning these table properties and show that it can consistently achieve superior performance over the competing baselines. Our theoretical and qualitative analyses also support the effectiveness of learning the structural properties.

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

# A  Details of the Proofs

## A.1  Preliminaries

**Definition A.1** (Hypergraph). Let $H = (V, E, \boldsymbol{X}, \boldsymbol{E})$ denote a hypergraph $H$ with a finite vertex set $V = \{v_1, \ldots, v_n\}$, corresponding vertex features with $\boldsymbol{X} \in \mathbb{R}^{n \times d}$; $d > 0$, a finite hyperedge set $E = \{e_1, \ldots, e_m\}$, where $E \subseteq P^*(V) \setminus \{\emptyset\}$ and $\bigcup\limits_{i=1}^{m} e_i = V$, and $P^*(V)$ denotes the power set on the vertices, the corresponding hyperedge features $\boldsymbol{E} \in \mathbb{R}^{m \times d}$; $d > 0$. The hypergraph connections can be conveniently represented as an incidence matrix $\mathbf{B} \in \{0, 1\}^{n \times m}$, where $\mathbf{B}_{ij} = 1$ when node $i$ belong to hyperedge $j$ and $\mathbf{B}_{ij} = 0$ otherwise.

**Definition A.2** (1-Wesifeiler Lehman Test). Let $G = (V, E)$ be a graph, with a finite vertex set $V$ and let $s : V \to \Delta$ be a node coloring function with an arbitrary co-domain $\Delta$ and $s(v), v \in V$ denotes the color of a node in the graph. Correspondingly, we say a labeled graph $(G, s)$ is a graph $G$ with a complete node coloring $s : V \to \Delta$. The 1-WL Algorithm [Babai et al., 1980] can then be described as follows: let $(G, s)$ be a labeled graph and in each iteration, $t \geq 0$, the 1-WL computes a node coloring $c_s^{(t)} : V \to \Delta$, which depends on the coloring from the previous iteration. The coloring of a node in iteration $t > 0$ is then computed as $c_s^{(t)}(v) = \text{HASH}\left(\left(c_s^{(t-1)}(v), \left\{c_s^{(t-1)}(u) \mid u \in N(v)\right\}\right)\right)$ where HASH is bijective map between the vertex set and $\Delta$, and $N(v)$ denotes the 1-hop neighborhood of node $v$ in the graph. The 1-WL algorithm terminates if the number of colors across two iterations does not change, i.e., the cardinalities of the images of $c_s^{(t-1)}$ and $c_s^{(t)}$ are equal. The 1-WL isomorphism test, is an isomorphism test, where the 1-WL algorithm is run in parallel on two graphs $G_1, G_2$ and the two graphs are deemed non-isomorphic if a different number of nodes are colored as $\kappa$ in $\Delta$.

**Definition A.3** (Group). A group is a set $G$ equipped with a binary operation $\cdot : G \times G \to G$ obeying the following axioms:

- for all $g_1, g_2 \in G$, $g_1 \cdot g_2 \in G$ (closure).
- for all $g_1, g_2, g_3 \in G$, $g_1 \cdot (g_2 \cdot g_3) = (g_1 \cdot g_2) \cdot g_3$ (associativity).
- there is a unique $e \in G$ such that $e \cdot g = g \cdot e = g$ for all $g \in G$ (identity).
- for all $g \in G$ there exists $g^{-1} \in G$ such that $g \cdot g^{-1} = g^{-1} \cdot g = e$ (inverse).

**Definition A.4** (Permutation Group). A permutation group (a.k.a. finite symmetric group) $\mathbb{S}_m$ is a discrete group $\mathcal{G}$ defined over a finite set of size $m$ symbols (w.l.o.g. we shall consider the set $\{1, 2, \ldots, m\}$) and consists of all the permutation operations that can be performed on the $m$ symbols. Since the total number of such permutation operations is $m!$ the order of $\mathbb{S}_m$ is m!.

**Definition A.5** (Product Group). Given two groups $G$ (with operation $+$) and $H$ (with operation $*$), the direct product group of $G \times H$ is defined as follows:

- The underlying set is the Cartesian product, $G \times H$. That is, the ordered pairs $(g, h)$, where $g \in G$ and $h \in H$
- The binary operation on $G \times H$ is defined component-wise: $(g1, h1) \cdot (g2, h2) = (g1 + g2, h1 * h2)$

**Definition A.6** (Group invariant functions). Let $G$ be a group acting on vector space $V$. We say that a function $f : V \to \mathbb{R}$ is $G$-invariant if $f(g \cdot x) = f(x) \quad \forall x \in V, g \in G$.

**Definition A.7** (Orbit of an element). For an element $x \in X$ and given a group $G$, the orbit of the element $x$ is given by $\text{Orb}(x) = \{g \cdot x \mid \forall g \in G\}$.

## A.2  Proofs

In the section, we restate (for the sake of convenience) and prove the statements in the paper.

First, we restate the assumption and the corresponding property and then provide the proof.

**Assumption A.8.** For any table $(\mathcal{T}_{ij})_{i \in [n], j \in [m]}$ (where $i, j$ are indexes of the rows, and columns appropriately), an arbitrary group action $g \in \mathbb{S}_n \times \mathbb{S}_m$ acting appropriately on the rows and columns leaves the target random variables associated with tasks on the entire table unchanged.

*Property* 2. A function $\phi : \mathcal{T} \mapsto \mathbf{z} \in R^d$ which satisfies Assumption A.8 and defined over tabular data must be invariant to actions from the (direct) product group $\mathbb{S}_n \times \mathbb{S}_m$ acting appropriately on the table i.e. $\phi(g \cdot \mathcal{T}) = \phi(\mathcal{T}) \ \ \forall g \in \mathbb{S}_n \times \mathbb{S}_m$.

*Proof.* Proof by contradiction. Let $\mathbf{y} \in \mathbb{R}$ be some target value associated with the entire table – which is invariant up to permutations of the table given by Assumption A.8. Assume the case where $\phi$ is not an invariant function acting over tables i.e. $\phi(g \cdot \mathcal{T}) \neq \phi(\mathcal{T}) \ \ \forall g \in \mathbb{S}_m \times \mathbb{S}_n$. This means different group actions will result in different output representations-and hence the representation is dependent on the action itself. When $d = 1$ and the value associated with $\mathbf{y}$ is given by $\phi$ acting over the table, equality between the two (for all permutation actions of a given) is only possible if we can unlearn the group action (or learn the same representation for the entire orbit) which would make it invariant–and hence a contradiction. $\qquad\square$

Next, we look at the function $g$ defined via Eqns (1-5) to see that the representation obtained by a table via message passing is invariant to row/column permutations of the incidence matrix $\mathbf{B}$. Eqn 1, which is the `Node2hyperedge` block, is a permutation invariant set aggregator which aggregates information from cells in the table to get the vectorial representation of a row/column/table hyperedge. That is, any permutation action drawn from the $\mathbb{S}_{mn}$ group doesn't impact the representation of a given hyperedge as long as the membership is consistent.

Similarly, the `Hyperedge2Node` block (Eqn 2) is an aggregation function that aggregates information from the hyperedges to the nodes. This ensures that a given node's (in the hypergraph) representation is invariant to permutation actions from the $\mathbb{S}_{m+n+1}$ group as long as the membership is consistent.

Now, since the `Node2Hyperedge` and `Hyperedge2Node` blocks act sequentially, its easy to see that (1-5) jointly encode invariance to the product group $\mathbb{S}_{mn} \times \mathbb{S}_{m+n+1}$ when we obtain representations of an incidence matrix as a set (permutation invariant) aggregation of representations of all the $mn$ nodes and the $m + n + 1$ hyperedges.

Next, we restate and prove the theorem about the maximal invariance property.

**Theorem A.9.** *A continuous function* $\phi : \mathcal{T} \mapsto \mathbf{z} \in \mathbb{R}^d$ *over tables is maximally invariant if it is modeled as a function* $g : \mathbf{B} \mapsto \mathbf{y} \in \mathbb{R}^k$ *over the incidence matrix of a hypergraph* $\mathcal{G}$ *constructed per Section 2.1 (Where $g$ is defined via Eqns (1-5)) if $\exists$ a bijective map between the space of tables and incidence matrices (defined over appropriate sizes of tables and incidence matrices). That is, $\phi(\mathcal{T}_1) = \phi(\mathcal{T}_2)$ iff $\mathcal{T}_2$ is some combination of row and/or column permutation of $\mathcal{T}_1$ and $g(\mathbf{B_1}) = g(\mathbf{B_2})$ where $\mathbf{B}_1, \mathbf{B}_2$ are the corresponding (hypergraph) incidence matrices of tables $\mathcal{T}_1, \mathcal{T}_2$.*

*Proof.* Proof by construction. We will assume the existence of a bijective map between the space of tables and incidence matrices (with appropriate indexing by column names and some canonical row ordering). Without loss of generality, we assume the number of rows and columns in the table to be $n$ and $m$ respectively, and some canonical ordering of the rows and columns in the table for the sake of convenience. Hence the number of nodes, and hyperedges in the hypergraph based on the formulation in Section 2.1 is $mn$ and $m + n + 1$ respectively. An invariant function over the incidence matrices means that function is invariant to any consistent relabelling of nodes and hyperedges in the hypergraph–hypergraph incidence matrix of size $mn \times (m + n + 1)$ and hence (upon node/hyperedge relabeling) invariant to the $\mathbb{S}_{mn} \times \mathbb{S}_{m+n+1}$ group. It is important to note that any row/column permutation of the table affects the ordering of both the rows and columns of the incidence matrix.

The existence of a bijective map between $\phi$ and $g$ guarantees that we know exactly which elements (cells, rows, columns, table) participate in which nodes/hyperedges and which don't. This is evident as a different permutation of the rows/columns would result in different incidence matrices and hence the above information can be deconstructed by obtaining the mappings between all permutations of a table and corresponding incidence matrices. Now, as this is proof of maximal invariance we have to prove both directions i.e. when the function $\phi$ maps two tables to the same representation-the tables are just row/column permutations of each other, and the $g$ over their corresponding incidence matrices yield identical representations and vice versa.

**Direction 1**: Assume a continuous function $\phi$ over two tables $\mathcal{T}_1, \mathcal{T}_2$ yields the same representation $\phi(\mathcal{T}_1) = \phi(\mathcal{T}_2)$. Now, we need to show that the function $g$ over the corresponding incidence matrices

$\mathbf{B}_1, \mathbf{B}_2$ yield the same representation. Let $\rho$ be the bijective map between the space of tables and incidence matrices. The guarantee of a bijective map between tables and incidence matrices and the injective (one-to-one) nature of AllSet [Chien et al., 2022] which is a multiset function that ensures that the hyperedges with the same elements get the same representation and non-identical hyperedges obtain different representations (via Eqns(1-5). Hence, when the outputs of $\phi$ match, the outputs of $g$ also do - and this is possible when the tables are identical up to permutations of each other. The universality property of AllSet, combined with the existence of the bijective map ensures the existence of a neural network $g$ for a given continuous function $\phi$.

**Direction 2**: Assume a function $g$ which maps incidence matrices $\mathbf{B}_1, \mathbf{B}_2$ of two hypergraphs to the same representation. Now again, since the bijective map exists, it ensures that we can check if the same nodes and hyperedges map across the two incidence matrices. This confirms the existence of a bijective map between the family of stars [Tyshkevich and Zverovich, 1996] of the two hypergraphs (i). The 1-Weisfeiler Lehman message passing equations of the `HyperTrans` (Along with the inherited injectivity property of AllSet) module along the star expansion of the hypergraph ensures that the star expansions of the hypergraphs are 1-WL isomorphic [Xu et al., 2018] (ii). Combining (i) and (ii) together via Theorem 2 from [Srinivasan et al., 2021], we get that the hypergraphs are isomorphic - and therefore yielding the tables to be identical and $\phi$ getting the same representation due to Property 1. It is easy to see that when the $g$ yields different representations, the tables are not identical, and hence $\phi$ yields a different representation as well. □

# B Further Analysis of HYTREL

## B.1 Effect of Input Table Size

As mentioned in Section 3, we have truncated the large tables and only keep a maximum of 30 rows and 20 columns in pretraining for efficiency purposes. However, the truncation does not affect HYTREL's ability to deal with large tables in downstream tasks. Our model can deal with the arbitrary size of tables. This is different from the BERT-based TaLMs (e.g., TaBERT, TURL) that have positional encodings, and the pretrained checkpoints limit the positions within a length of 512 for an input sequence. With the HYTREL, we can always build large hypergraphs with large tables by adding more nodes under each hyperedge, without sacrificing information loss.

To further demonstrate this, we do experiments on the table type detection (TTD) dataset, as it has relatively large tables (an average number of 157 rows). Below in Table 3 are the experimental results of using different maximum row/column limitations with HYTREL (ELECTRA), and we fine-tune the dataset on an NVIDIA A10 Tensor Core GPU (24GB).

| Size Limitations (#rows, #columns) | Dev Acc (%) | Test Acc (%) | Training Time (min/epoch) | GPU Memory Consumption (memory used/batch size, GB/sample) |
|---|---|---|---|---|
| (3, 2) | 78.47 | 78.00 | 3 | 0.14 |
| (15, 10) | 95.36 | 95.96 | 9 | 0.48 |
| (30, 20) | 96.23 | 95.81 | 14 | 0.58 |
| (60, 40) | 96.38 | 95.80 | 23 | 1.41 |
| (120, 80) | 96.02 | 95.71 | 51 | 5.13 |
| (240, 160) | 96.06 | 95.67 | 90 | 16.00 |

Table 3: Effect of Input Table Size

We can see that HYTREL has no difficulty dealing with the arbitrary size of input tables. However, we also observe that the prediction performances are almost the same when we enlarge the table size limitation to a certain number, but the time and hardware resource consumption will increase dramatically. So we recommend using downsampling to deal with large tables, and it will be much more efficient without hurting the performance.

## B.2 Effect of Excessive Invariance

As we have emphasized, the invariant permutation property of a table restricts the permutation of rows and columns to be independent. Arbitrary permutations of rows or columns can lead to an

excessive invariance problem, ultimately compromising the table's semantics. Take Figure 6 as an example, permuting the rows and columns independently will result in semantically the same table (Upper), but arbitrarily permuting will break the semantics of the table and cause excessive invariance (Below).

We use the TaBERT model to demonstrate the effect brought by the excessive invariance. As we have introduced before, the TaBERT models use a linearization approach with positional encoding to model a table. So they cannot preserve the table invariance property. However, we cannot achieve this by simply removing the positions, as it will cause excessive invariance and it is equivalent to shuffling all the cells in a table. As we emphasize, we only assume independent row/column permutation invariance. We empirically show the effect of disabling the positional encodings for the TaBERT model. As shown in Table 5, we evaluate the model on the Table Type Detection dataset. It turns out that the excessive invariance will cause a significant performance decrease.

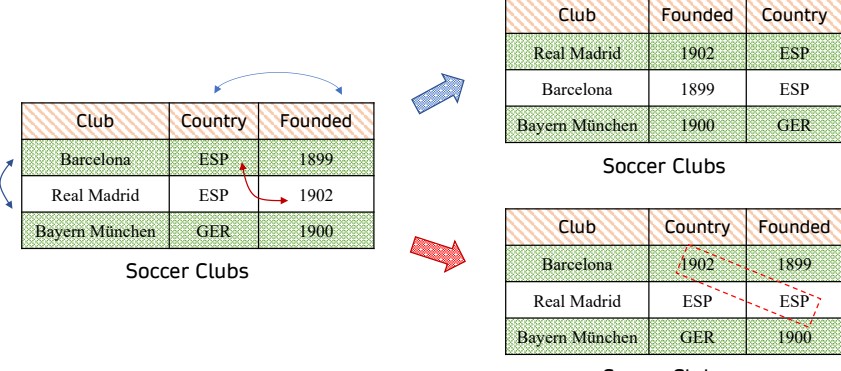

Figure 6: An example of table permutations.

| Models | Dev Acc (%) | Test Acc (%) |
|---|---|---|
| TaBERT (K=1), w/ positions | 94.11 | 93.44 |
| TaBERT (K=1), w/o positions | 88.62 | 88.91 |
| TaBERT (K=3), w/ positions | 95.82 | 95.22 |
| TaBERT (K=3), w/o positions | 92.80 | 92.44 |

Table 4: Excessive invariance causes performance slumping

### B.3 Inference Complexity of the Model

**Theoretical Analysis**: Given a table with $m$ rows and $n$ columns, let's assume each cell only contains one token, and we use dot-product and single head in the attention operation. The inference complexity comparison of one layer attention is:

| Complexity | BERT-based Methods | HYTREL |
|---|---|---|
| w/o linear transformation | $O((mn)^2 \cdot d)$ | $O(mnd)$ |
| w/ linear transformation | $O((mn)^2 \cdot d + (mn) \cdot d^2)$ | $O(mnd + (mn) \cdot d^2)$ |

Table 5: Inference Complexity

For the BERT-based methods, the input sequence length is $mn$, $d$ is the hidden dimension. We can see that our HyTrel has less complexity, and the difference comes from the dot-product for attention scores. The query in the set transformer of HyTrel is a learnable vector with $d$ dimension, it reduces computation time of self-attention from quadratic to linear in the number of elements in the set [Lee et al., 2019], with a complexity of $O(mnd)$. In the self-attention, the query is a matrix from the

whole input sequence with $(mn) \times d$ dimensions, which has a complexity of $O((mn)^2 \cdot d)$ in the attention score calculation.

**Empirical Analysis**: We include comparison of our approach with the TaBERT baseline about the inference time, including data processing (graph building time in HyTrel). As observable from the table, the total inference time of our model is lesser compared to TaBERT, and the time cost for graph building is not a bottleneck.

| Datasets | CTA | CPA | TTD | TS |
|---|---|---|---|---|
| Total Samples | 4844 | 1560 | 4500 | 1391 |
| batch size | 8 | 8 | 8 | 8 |
| TaBERT (K=1) - data preparing | 13 | 5 | 51 | 13 |
| TaBERT (K=1) - inference | 27 | 5 | 25 | 28 |
| **TaBERT (K=1) - total inference** | **40** | **10** | **76** | **41** |
| TaBERT (K=3) - data preparing | 14 | 5 | 52 | 13 |
| TaBERT (K=3) - inference | 85 | 16 | 79 | 80 |
| **TaBERT (K=3) - total inference** | **99** | **21** | **131** | **93** |
| HyTrel - graph building | 5 | 1 | 16 | 5 |
| HyTrel - inference | 16 | 6 | 17 | 13 |
| **HyTrel - total inference** | **21** | **7** | **33** | **18** |

Table 6: Inference Time Comparision (seconds)

- We keeps a maximal of 3 rows with the HyTrel model for fair comparison with the BERT (K=3) models.
- The data preprocessing of TaBERT is to format the input tables (.json) into tensors, and the graph building of HyTrel is to format the input tables (.json) into feature tensors and the incidence matrix in hypergraphs.
- All experiments are conducted on a single A10 GPU, and the inference batch sizes are all chosen to be 8 for all models and all dataset;
- We use the validation set of CTA, CPA and TTD for experiments. For TS with a small number of tables that is tested with five fold cross-validation in the paper, here we use the whole dataset for experiments.

## C  Details of the Experiments

### C.1  Pretraining

#### C.1.1  Model Size

The parameter size of the HYTREL and the baseline TaBERT are as followed in Table 7. HYTREL has more parameters mainly because of the `Hyperedge Fusion` block (Figure 2) that is absent from the TaBERT model.

| Models | # Parameters (million) |
|---|---|
| TaBERT (K=1) | 110.6 |
| TaBERT (K=3) | 131.9 |
| HYTREL | 179.4 |

Table 7: Model Size

#### C.1.2  Hyperparameters

The hyperparameter choices and resource consumption details of the pretraining are presented in Table 8 and Table 9.

| Models | Batch Size | Learning Rate | Warm-up Ratio | Max Epochs | Weight Decay | Optimizer |
|---|---|---|---|---|---|---|
| HYTREL *w/* ELECTRA | 8192 | $5e^{-4}$ | 0.05 | 5 | 0.02 | Adam |
| HYTREL *w/* Contrastive | 4096 | $1e^{-4}$ | 0.05 | 5 | 0.02 | Adam |

Table 8: Pretraining Hyperparameters

| Models | GPUs | Accelerator | Precision | Training Time |
|---|---|---|---|---|
| HYTREL *w/* ELECTRA | 16 × NVIDIA A100 | DeepSpeed (ZeRO Stage 1) | bf16 | 6h / epoch |
| HYTREL *w/* Contrastive | 16 × NVIDIA A100 | DeepSpeed (ZeRO Stage 1) | bf16 | 4h / epoch |

Table 9: Pretraining Cost

## C.2 Fine-tuning

### C.2.1 Details of the Datasets

**TURL-CTA** is a multi-label classification task (this means for one column, there can be multiple labels annotated) that consists of 255 semantic types. In total, it has 628,254 columns from 397,098 tables for training, and 13,025 (13,391) columns from 4,764 (4,844) tables for testing (validation).

**TURL-CPA** dataset is also a multi-label classification task in which each pair of columns may belong to multiple relations. It consists of a total number of 121 relations with 62,954 column pairs from 52,943 tables training sample, and 2,072 (2,175) column pairs from 1,467 (1,560) tables for testing (validation). Similar to the CTA task, we use the pairwise column representations from HYTREL with their corresponding hyperedge representations for fine-tuning.

**WDC Schema.org Table Corpus** consists of 4.2 million relational tables covering 43 schema.org entity classes. We sample tables from the top 10 classes and construct a table type detection dataset that contains 3,6000 training tables and 4,500 (4,500) tables for validation (testing).

**PMC** is a table corpus that is formed from PubMed Central (PMC) Open Access subset, and this collection is about biomedicine and life sciences. In total, PMC contains 1,391 table pairs, where 542 pairs are similar and 849 pairs are dissimilar. For each pair of tables, we use their table representations together with column representations from HYTREL for fine-tuning. As for evaluation, we follow previous work [Habibi et al., 2020, Trabelsi et al., 2022] and report the macro-average of the 5-fold cross-validation performance.

### C.2.2 Hyperparameters

The hyperparameter choices for fine-tuning both the TaBERT and the HYTREL models are as followed. For the primary hyperparameters `batch size` and `learning rate`, we choose the `batch size` to be as large as possible to fit the GPU memory constraint, and then use grid search to find the best `learning rate`. Other hyperparameters are kept the same as in previous work.

| Tasks | Batch Size | Learning Rate | Warm-up Ratio | Max Epochs | Weight Decay | Optimizer |
|---|---|---|---|---|---|---|
| Column Type Annotation | 256 | $1e^{-3}$ | 0.05 | 50 | 0.02 | Adam |
| Column Property Annotation | 256 | $1e^{-3}$ | 0.05 | 30 | 0.02 | Adam |
| Table Type Detection | 32 | $5e^{-5}$ | 0.05 | 10 | 0.02 | Adam |
| Table Similarity Prediction | 16 | $5e^{-5}$ | 0.05 | 10 | 0.02 | Adam |

Table 10: Hyperparameters for Fine-tuning HYTREL. We use the same settings for all variants of HYTREL (no pretraining, pretraining with ELECTRA and Contrastive objectives)

| Tasks | Batch Size | Learning Rate | Warm-up Ratio | Max Epochs | Optimizer |
|---|---|---|---|---|---|
| Column Type Annotation | 64 (K=3), 128 (K=1) | $5e^{-5}$ | 0.1 | 10 | Adam |
| Column Property Annotation | 48 (K=3), 128 (K=1) | $1e^{-5}$ (K=3), $2e^{-5}$ (K=1) | 0.1 | 10 | Adam |
| Table Type Detection | 64 | $5e^{-5}$ | 0.1 | 10 | Adam |
| Table Similarity Prediction | 8 | $5e^{-5}$ | 0.1 | 10 | Adam |

Table 11: Hyperparameters for Fine-tuning TaBERT.

### C.2.3 Details of the Baselines

**TaBERT** [Yin et al., 2020] is a representative TaLM that jointly learns representations for sentences and tables. It flattens the tables into sequences and pretrains the model from BERT checkpoints. Since we use the same source and prepossessing for the pretraining tables, we use it as our main baseline. TaBERT is not yet evaluated on the table-only tasks, we fine-tune the *base* version with both *K=1* and *K=3* row number settings on all the tasks. We also evaluate the TaBERT model that is randomly initialized and see how much it relies on the pretraining.

**TURL** [Deng et al., 2020] is another representative TaLMs that focuses on web tables with cell values as entities connected to the knowledge base. It also flattens the tables into sequences and pretrains from TinyBERT [Jiao et al., 2020] checkpoints. It introduces a vision matrix to incorporate table structure into the representations. We copy the reported results for CTA and CPA tasks. For a fair comparison, we use the fine-tuning settings with metadata (captions and headers) but without external pretrained entity embeddings.

**Doduo** [Suhara et al., 2022] is a state-of-the-art column annotation system that fine-tunes the BERT and uses table serialization to incorporate table content. We copy the reported results for the CTA and CPA tasks.

For each specific task and dataset, we also report the state-of-the-art baseline results from previous work. For the TURL-CTA dataset, we use the reported results of Sherlock [Hulsebos et al., 2019]. For the TURL-CPA dataset, we use the reported results of fine-tuning the $BERT_{base}$ [Devlin et al., 2019] model. For the PMC dataset, we use the reported results from two baselines: TFIDF+Glove+MLP, which uses TFIDF and Glove [Pennington et al., 2014] features of a table with a fully connected MLP model, and TabSim [Habibi et al., 2020] that uses a siamese Neural Network to encode a pair of tables for prediction.

