# OpenReview forum: "HyTrel: Hypergraph-enhanced  Tabular Data Representation Learning"
_NeurIPS.cc/2023/Conference — NeurIPS 2023 spotlight_

### Official Review · Reviewer_SuH5 · 2023-06-26

**Soundness:** 3 good
**Presentation:** 3 good
**Contribution:** 3 good
**Rating:** 6
**Confidence:** 4

**Summary:**

This paper proposes to transform a source table into a hypergraph. Each cell is a node in the graph, where nodes in the same row, column, and table are connected using three types of hyperedges. The authors claimed that by modeling the hypergraph of the table, the proposed method, named *HyTrel*, is able to generate representations of tables robust to permutations. Two self-supervised objectives: (1) cell or header corruption indication prediction and (2) in-batch negative sampling with InfoNCE contrastive loss for paired nodes.

The experiments were performed on (1) pre-training on 27M tables and (2) fine-tuning for four tabular understanding tasks: column type annotation, column property annotation, table type detection, and table similarity prediction. The empirical findings include

- Pre-training on HyTrel leads to marginal improvement over the variant without pre-training.

- No clear evidence of which pre-training objectives are better, considering the four tasks evaluated.

**Strengths:**

- It is reasonable to seek permutation invariance in tabular data representation as the existing table serialization approach fails to achieve this property.

- The proposed method surpasses previous methods by utilizing directly supervised training on the target data without pre-training.

**Weaknesses:**

**Complexity**
The main concern lies in the efficiency of the proposed method. It needs to build a huge hypergraph on the tables, where the number of nodes is equal to the number of cells in the table, and the number of edges explodes when we combine nodes, columns, and rows. As we know, it is very common that a table may contain millions of rows and thousands of columns. However, there seems to be lacking complexity analysis of the overall running time and memory cost.

Although HyTrel was observed converging faster than previous BERT-based methods in terms of the number of epochs, there is no direct comparison of the inference time needed for each sample/batch. Also, the time required for building the graph in the training and the prediction phase should be taken into account.

**Questions:**

Please refer to the weaknesses section.

**Limitations:**

Please refer to the weaknesses section.

---

> ### Author Rebuttal · Authors · 2023-08-10
>
> **Q1**: Complexity. The main concern lies in the efficiency of the proposed method. It needs to build a huge hypergraph on the tables, where the number of nodes is equal to the number of cells in the table, and the number of edges explodes when we combine nodes, columns, and rows. As we know, it is very common that a table may contain millions of rows and thousands of columns. However, there seems to be lacking complexity analysis of the overall running time and memory cost.
>
> **A1**: a. Given a table with $m$ rows and $n$ columns, the corresponding hypergraph will have $mn$ cells, and $m+n+1$ hyperedges. So the number of hyperedge is **linearly** related to the table size.
>
> b. As for the large table input, HyTrel can deal with **arbitrary size of tables theoretically**, because the HyperTrans model structure does not have positional encodings that constrain the sequence length as in common transformer models. **We have included such analysis and the overall running time and memory cost for relatively large tables in our submitted appendix** (Appendix B1, Table 3), and we also include it here for your reference.
>
> |Size Limitations (#rows, #columns) | Dev Acc (%) | Test Acc (%) | Training Time (min/epoch) | GPU Memory Consumption (memory used/batch size, GB/sample) |
> |-|-|-|-|-|
> |(30, 20) |	96.23 |	95.81	|14	|0.58|
> |(60, 40) |	96.38 | 95.80	| 23	| 1.41 |
> |(120, 80)|	96.02 |	95.71 |	51	| 5.13 |
> |(240, 160) |	96.06 |	95.67 |	90 |	16.00 |
>
> When dealing with large tables, downsampling of rows and columns to represent the table content is a commonly used practice. As shown in the above table, we use the table type detection dataset for experiment as it contains relatively large tables. We sample a subset of rows and columns of a large table by limiting the maximal rows and columns count (we truncate tail rows and columns).
>
> We observe that **appropriate downsampling does not hurt the performance** for table level tasks (and we hypothesize the same for column-level prediction tasks). We note that, as we sample more rows, the performance increases. However, the performance soon saturates but incurs a huge increase in time and memory consumption. So in our evaluation, we limit to using just 30 rows and 20 columns, as HyTrel can already perform close to its peak with reasonable time and memory cost.
>
>
>
>
>
> **Q2**. Although HyTrel was observed converging faster than previous BERT-based methods in terms of the number of epochs, there is no direct comparison of the inference time needed for each sample/batch. Also, the time required for building the graph in the training and the prediction phase should be taken into account.
>
> **A2**: a. Theoretical Analysis: given a table with $m$ rows and $n$ columns, let’s assume each cell only contains one token, and we use dot-product and single head in the attention operation. The inference complexity comparison of one layer attention is:
> ||BERT-based methods | HyTrel |
> |-|-|-|
> |w/o linear transformation | $O((mn)^2 \cdot d)$| $O((mn)^2)$|
> |w/ linear transformation | $O((mn)^2 \cdot d) + (mn)d^2)$| $O((mn)^2) +(mn)d^2)$|
>
> For the BERT-based methods, the input sequence length is $mn$. $d$ is the hidden dimension.
>
> We can see that our HyTrel has less complexity, and the difference comes from the dot-product for attention scores. The query in the set transformer of HyTrel is a learnable vector with $d$ dimension, then the bottleneck comes from the softmax calculation with a complexity of $O((mn)^2)$. In the self-attention, the query is a matrix from the whole input sequence with $mn \times d$ dimensions, which has a complexity of $O((mn)^2 \cdot d)$ in the attention score calculation.
>
> b. Empirical Analysis
> | Time (seconds) | CTA (4844 samples) | CPA (1560 samples) | TTD (4500 samples) | TS (1391 samples) |
> | - | - | - | - | - |
> |batch size | 8 | 8 | 8 | 8 |
> |TaBERT (K=1) - data preparing | 13 | 5 | 51 | 13 |
> |TaBERT (K=1) - inference | 27 | 5 | 25 | 28 |
> | **TaBERT (K=1) - total inference** | **40**  | **10** | **76**  | **41** |
> |TaBERT (K=3) - data preparing | 14 | 5 | 52 | 13 |
> |TaBERT (K=3) - inference | 85 | 16 | 79 | 80 |
> | **TaBERT (K=3) - total inference** | **99**  | **21** | **131**  | **93** |
> |HyTrel - graph building | 5 | 1 | 16 | 5 |
> |HyTrel - inference | 16 | 6 | 17 | 13 |
> | **HyTrel - total inference** | **21**  | **7** | **33**  | **18** |
>
> We include comparison of our approach with the TaBERT baseline about the inference time, including data processing (graph building time in HyTrel). As observable from the table, **the total inference time of our model is lesser compared to TaBERT, and the time cost for graph building is not a bottleneck**.
>
> Note:
> 1. We keeps a maximal of 3 rows with the HyTrel model for fair comparison with the BERT (K=3) models.
> 2. The data preprocessing of TaBERT is to format the input tables (.json) into tensors, and the graph building of HyTrel is to format the input tables (.json) into feature tensors and the incidence matrix in hypergraphs.
> 3. All experiments are conducted on a single A10 GPU, and the inference batch sizes are all chosen to be 8 for all models and all dataset;
> 4. We use the validation set of CTA, CPA and TTD for experiments. For TS with a small number of tables that is tested with five fold cross-validation in the paper, here we use the whole dataset for experiments.

---

> ### Comment · Reviewer_SuH5 · 2023-08-19
>
> Thanks for your response and I will update my score accordingly.

---

> > ### Author Response · Authors · 2023-08-20
> >
> > Thank you for revisiting our work and considering our response; we appreciate your willingness to update the score.

---

### Official Review · Reviewer_4rho · 2023-07-03

**Soundness:** 3 good
**Presentation:** 3 good
**Contribution:** 3 good
**Rating:** 7
**Confidence:** 3

**Summary:**

The paper proposes a framework for tabular representation learning by modeling the structure of the tables as a hyper-graph working on different granularities, namely, rows, columns, and the entire table.

**Strengths:**

- The paper addresses a substantial question of how to best incorporate the structure of a table while pre-training models for tabular data.
- The proposed framework outperforms the baselines on tasks purely dependent on the learned table representations.
- The paper is well-written and easy to follow.
- In addition, the authors also include quantitative analysis to demonstrate that their framework can assimilate the table structures to generate robust representations.


**Weaknesses:**

There have been recent works proposing techniques to incorporate the table structure while pre-training or fine-tuning (by biasing the attention layer or tasks other than the ones considered in the paper). It will strengthen the paper if the authors discuss how their work differs from those.

- MATE: Multi-view Attention for Table Transformer Efficiency Julian Martin Eisenschlos, Maharshi Gor, Thomas Müller, William W. Cohen EMNLP 2021
- Learning Enhanced Representations for Tabular Data via Neighborhood Propagation Kounianhua Du, Weinan Zhang, Ruiwen Zhou, Yangkun Wang, Xilong Zhao, Jiarui Jin, Quan Gan, Zheng Zhang, David Wipf NeurIPS 2022
- Learning Representations without Compositional Assumptions Tennison Liu, Jeroen Berrevoets, Zhaozhi Qian, Mihaela van der Schaar ICML 2023

**Questions:**

- Does the model also include a classification head like other TaLMs? If not, why did the authors not consider including it? This simple change would have opened up evaluation for some more benchmarks.

- What's the rationale behind the modified ELECTRA and contrastive objective for pre-training (as opposed to other objectives)?

- How would the approach perform if tables have a lot of numerical values? Will numeric tables lead to a degradation in performance?

**Limitations:**

The authors include a section in the paper that goes through the relevant limitations of the proposed approach.

---

> ### Author Rebuttal · Authors · 2023-08-10
>
> **Q1**: Missing Inferences
>
> **A1**: Thanks for pointing out and we will include them in our paper. Here are the relevance and differences of these papers compared with ours.
>
> [1] MATE belongs to the second group of studies we have categorized in Section 5 (Related Work). MATE explicitly restricts the attention of the transformer models to be applied within the same row and columns, and the sparse attention mechanism can be more efficient and it also learn spart of the table structures. **This paradigm is very similar to our baseline approach TURL**. However, our hypergraph-enhanced approach can maximally take advantage of the four table structures as we have emphasized in the paper, and our HyperAtt block with set transformers is also efficient as it does not need to pairwise calculate the attention scores among all the cells.
>
> [2] NeurIPS'22 is **already included in our related work (line 305)**. It belongs to the first group of studies that focus on predicting labels (essentially one row) for classification and regression problems, using row information and column schema as input. While the method used in this paper uses hypergraphs to model table structure, the problem solved in this paper is fundamentally different from ours. Our work belongs to the second group of study (Tabular language models) that focuses on using the hypergraphs to obtain task agnostic representations as part of pre-training.
>
> [3] ICML'23 is a contemporary work (**public after our submission**) that belongs to the first group of studies that focus on predicting row labels, like [2]. It focuses on the scenario with multi-views (different sets of features for an object) for a given table. It uses graph auto-encoder models to enforce information propagation among different views. The target tasks and motivations to use graphs in the paper are significantly different from ours.
>
> **Q2**: Does the model also include a classification head like other TaLMs? If not, why did the authors not consider including it? This simple change would have opened up evaluation for some more benchmarks.
>
> **A2**: If we understand you correctly, the classification head you mentioned is the special tokens that represent the special elements of a table, like the ‘[CLS]’ token from the BERT and TaBERT that represents the whole sentence or the whole table. In our settings, we use the last-layer node representations (each corresponds to a cell)  and the hyperedge representations (each corresponds to a row, a column or the whole table) as an input for the classification head. **We can use them for any relevant downstream tasks**.
>
> **Q3**: What's the rationale behind the modified ELECTRA and contrastive objective for pre-training (as opposed to other objectives)?
>
> **A3**: The rationale behind the modified ELECTRA is in line with the motivation of masked token prediction in the Word2Vec and the BERT model. In our case, if the context of a cell (the representations of the target cell is aggregated from its surroundings)  in a table can predict the value of a cell, it means the representations of the context ‘knows’ the cell. Then it can incorporate the information from predicted cells during the pretraining. However, the prediction space is extremely large for cell values as compared with the vocabulary size in the language model. Inspired by the original ELECTRA and [4], an alternative efficient approach is just to predict whether the cell has been corrupted or not. Our empirical results are consistent with previous work, demonstrating the effectiveness of this strategy.
>
> As for the rationale behind the contrastive objective, it is inspired by the contrastive learning over hypergraphs [5]. This work has proposed effective ways to augment hypergraphs for contrastive learning, and learn good representations for the nodes and hyperedges. Since we find it beneficial to model a table as a hypergraph for table structures, we also tried the contrastive over hypergraphs to learn the table representation. An intuitive rationale behind this is that similar elements of a table should be grouped together, and dissimilar ones should be separated. Our empirical results further demonstrate its effectiveness.
>
> **Other objective**: We also tried to reconstruct the incidence matrix of the hypergraph as a part of the objective (inspired by graph auto-encoder)  -  however this does not yield any performance gain.
>
>
> [4] Iida, Hiroshi, et al. "Tabbie: Pretrained representations of tabular data." arXiv preprint arXiv:2105.02584 (2021).
>
> [5] Wei, Tianxin, et al. "Augmentations in hypergraph contrastive learning: Fabricated and generative." Advances in neural information processing systems 35 (2022): 1909-1922.
>
> **Q4**: How would the approach perform if tables have a lot of numerical values? Will numeric tables lead to a degradation in performance?
>
> **A4**: We use numBERT [6] to deal with numerical values, which turns all numerical value into scientific format with a special token "scinotexp". To demonstrate whether numeric tables will lead to degradation in performance, we experiment with the TTD benchmark as it contains tables that have rich numerical values. We divide the test set into two subsets: one subset with a large fraction of numerical cell values (>40%) and one with a smaller fraction (<40%). We test them by fine-tuning the HyTrel ELECTRA version both with numBERT preprocessing and without it. The results demonstrate that **numerical tables tend to degrade the performance by 2.23%, and numBERT can help alleviate the gap ( from 2.23 to 1.58)**. It demonstrates that our approach can take advantage of the numBERT.
>
>
> |TTD (accuracy, %) | numerical values > 40 %| numerical values < 40 % | difference |
> |-|-|-|-|
> |HyTrel - ELECTRA | 94.65 | 96.23 |1.58 |
> |HyTrel - ELECTRA w/o numBERT | 93.68 | 95.91 |2.23|
>
>
> [6] Zhang, X., Ramachandran, D., Tenney, I., Elazar, Y., & Roth, D. (2020). Do language embeddings capture scales?. arXiv preprint arXiv:2010.05345.

---

> > ### Comment · Reviewer_4rho · 2023-08-14
> >
> > Thank you for your detailed response. I appreciate it.

---

> > > ### Author Response · Authors · 2023-08-14
> > >
> > > Thank you for your consideration and the time you've taken to review our work. We appreciate your feedback.

---

### Official Review · Reviewer_7wTp · 2023-07-04

**Soundness:** 3 good
**Presentation:** 3 good
**Contribution:** 3 good
**Rating:** 7
**Confidence:** 3

**Summary:**

This work presents a novel tabular language model that represents tables as hypergraphs called HyTrel. HyTrel is designed to capture the structural properties of tabular data, including 1) invariance to row/column permutations, 2) structural similarity within columns, 3) high-order multilateral relations, and 4) hierarchical organization. However, many of the Language models pre-trained on tabular data do not consider these structural properties that exist in tabular data. Experimental results are provided to demonstrate the superiority of the HyTrel on four downstream tasks, and the robustness of the representations from HyTrel is further validated through qualitative analysis.

**Strengths:**

1. Unlike previous methods, the HyTrel model can handle changes in the order of rows and columns without significantly affecting its performance.
2. Also, HyTrel can handle input tables of arbitrary size, making it versatile and adaptable to a wide range of datasets.
3. The model is highly efficient regarding the number of epochs for pretraining compared to prior works.
4. HyTrel can incorporate the hierarchical table structure into its representations.
5. By incorporating these structural properties, HyTrel consistently achieves superior performance over the competing baselines.

**Weaknesses:**

No detailed comparison of computational efficiency or runtime on the number of epochs needed for pretraining compared to prior works.

**Questions:**

Compared to the previous method, how would HyTrel representation of Tabular Data benefit Multilingual Tabular data alignment?

**Limitations:**

HyTrel is designed for tables with simple column structures and struggles with tables that have complex hierarchical column structures.

---

> ### Author Rebuttal · Authors · 2023-08-10
>
> **Q1**: No detailed comparison of computational efficiency or runtime on the number of epochs needed for pretraining compared to prior works.
>
> **A1**: Thanks for pointing out the pretraining comparison. We **have discussed the pretraining epochs** as compared with previous work in Section 4.3 (line 295 - 296). Here we include the detailed analysis for the pretraining cost as compared with our baseline TaBERT (the TURL paper does not report more pretraining cost details like the pretraining time and machine used, it only includes the pretraining epochs we have compared in the paper).
>
> As we can observe, our HyTrel models  are way more efficient (**480/320 v.s. 17,280 GPU hours, 5 v.s. 10 epochs**)
>
> || Backbone model | # Paras (million) | Pretraing corpus | Time cost per epoch | # epochs | GPUs used | Total time cost | Total GPU hours |
> |-|-|-|-|-|-|-|-|-|
> | HyTrel - ELECTRA | Random initialization | 179.4 | CommonCrawl + Wikipedia Tables | 6h | 5 | 16*A100 | 30h | 480 |
> | HyTrel - Contrast | Random initialization | 179.4 | CommonCrawl + Wikipedia Tables | 4h | 5 | 16*A100 | 20h | 320 |
> | TaBERT (K = 3) -Large | BERT-large | 340 | CommonCrawl + Wikipedia Tables | None|10 | 120*V100 | 6 days |17,280 |
>
> Please also note that:
>
> a. All the list model are pretrained with the same corpus (CommonCrawl WDC Web Table + Wikipedia Tables).
>
> b. The computation efficiency difference between the V100 and A100 is trivial as compared with the scale of the GPU hours (A100 is 2.6x faster than the V100 using mixed precision in language model pretraining, from lambdalab.com)
>
> c. The TaBERT paper only report the pretraining cost for the large version TaBERT, which has twice amount of parameters than our HyTrel models. However, the pretraing overheads are way more than twice than ours.
>
> d. Our models are pretrained from scratch (randomly initialized), while the TaBERT are pretrained from the checkpoints of BERT models. This means to get a ready model, the pretraining overheads are much more than the current GPU hours in the table for the baselines models .
>
> **Q2**: Compared to the previous method, how would HyTrel representation of Tabular Data benefit Multilingual Tabular data alignment?
>
> **A2**: Thanks for bringing up the attention to extend HyTrel to multilingual tabular data. From our understanding, multilingual representation alignment is a challenging problem, especially in the case of contextual representations [1]. To the best of our knowledge, we are not aware of significant studies of TaLMs that work on multilingual data alignment. However, the BERT-based TaLMs like TaBERT can always adopt the alignment textual representations alignment approaches like the multilingual BERT pretrained on 104 languages. Based on our HyTrel that models a table as a hypergraph, one possible approach to do the multilingual alignment is to pretrain parallel tables with a contrastive objective that aligns the parallel cells/rows/columns/tables from different languages. **One benefit we can expect from this approach is the extremely pretraining efficiency** in comparison with the multilingual pretraining approach from the BERT models, as we have discussed in the previous question.
>
> [1] Cao, S., Kitaev, N., & Klein, D. (2020). Multilingual alignment of contextual word representations. arXiv preprint arXiv:2002.03518.
>
> **Q3**: Limitation: HyTrel is designed for tables with simple column structures and struggles with tables that have complex hierarchical column structures.
>
> **A3**: Thanks for pointing out the limitation of dealing with complex hierarchical column structures. We have included such a limitation discussion in our paper (**section 6, line 343-345**). Even though our Hytrel is not explicitly designed for complex hierarchical columns structures, our model can still deal with such tables. As in the following example, we can **collapse the hierarchical headers into a simple structure, and then model the table as a hypergraph**. However, we leave special designs for such complex hierarchical column structures as future work.
>
>
> |Variable | Type of job contract | |
> |-|-|-|
> ||Informal (%) | Formal (%) |
> |Retail | 27.3 | 72.7 |
> |Services | 32.1 | 67.9 |
>
> |Variable | Type of job contract- Informal (%) | Type of job contract- Formal (%) |
> |-|-|-|
> |Retail | 27.3 | 72.7 |
> |Services | 32.1 | 67.9 |

---

> > ### Comment · Reviewer_7wTp · 2023-08-14
> > **Comments after Rebuttal**
> >
> > Thank you for your detailed response in addressing my concerns.  For this reason, I am willing to revise my score up.

---

> > > ### Author Response · Authors · 2023-08-14
> > >
> > > Thank you for revisiting our work and considering our response; we appreciate your willingness to revise the score.

---

### Official Review · Reviewer_kUWt · 2023-07-07

**Soundness:** 3 good
**Presentation:** 3 good
**Contribution:** 3 good
**Rating:** 6
**Confidence:** 3

**Summary:**

This paper proposes a tabular language model (HyTrel) that utilizes the hypergraphs of the data table. Specifically, the hypergraph is constructed with cell values representing each node and the row, column, table representing the hyeredges. In the proposed framwork, the nodes and edges are first fed into the embedding layer, followed by 12 layers of HyperTrans structure, which is consisted of two HyperAtt blocks connected by  a hyperedge fusion block. The model is pretrained through corruption of certain parts of the table/hypergraph. To use this model, one can first compute the table encoder of a table, which can then be used to finetune downstream tasks. This paper reports competitive performance of HyTrel in 4 downstream tasks (CTA, CPA, TTD, TSP).

**Strengths:**

The HyTrel architecture produces a table encoder that is permutation invariant with both theoretical supports and empirical experiments. It analyzes different functions in the HyperTrans layer and shows the output encoding will not be affected by row/column permutation of the table. Tasks finetuned with such encoder outperform other baseline models. There are also some qualitative analysis of the hierarchical structure of the table being learned through the proposed framework.

**Weaknesses:**

The method only applies to a single table scenario. In real-world applications, this could be a rare case where tabular learning is involved. Another dimension that can be added to the hypergraph can be key/foreign key relationships, which may extend the model to work in multi-table settings.

**Questions:**

What are some examples of realistic permutation of the table other than simple row/column permutation? If there are other non-trivial permutation types that can benefit from the invariance brought by HyTrel, I think it adds additional value and persuasion to the adopt this method from a practical point of view. Or, in the case where row/column permutation is the main target, it would be helpful to show how it negative affects downstream tasks, motivating the development of an permutation invariant encoding representation.

**Limitations:**

See the weakness and questions section above.

---

> ### Author Rebuttal · Authors · 2023-08-10
>
> **Q1**: The method only applies to a single table scenario. In real-world applications, this could be a rare case where tabular learning is involved. Another dimension that can be added to the hypergraph can be key/foreign key relationships, which may extend the model to work in multi-table settings.
>
> **A1**: Thanks for pointing out modeling the cross-table relationships. We agree with this point and we have such a discussion in our Limitation Section (**section 6, line 345-347**). Our work currently focuses on learning representations for single tables, which is still a challenging and important problem for table understanding going beyond our table evaluation tasks. We believe that our hypergraph approach can generalize to model the cross-table interactions via key/foreign key relationship, and we leave these aspects for future research.
>
> **Q2**: What are some examples of realistic permutation of the table other than simple row/column permutation? If there are other non-trivial permutation types that can benefit from the invariance brought by HyTrel, I think it adds additional value and persuasion to the adopt this method from a practical point of view.
>
> **A2**: To the best of our knowledge, we are not aware of any other non-trivial permutation types that exist in general tables. The arbitrary row and column permutations (independently) that we model are in line with real-world situations. For example, insertion/deletion in arbitrary positions on tables are common database operations which result in the  order of rows and columns being permuted. **This necessitates the need for learning table representations that are invariant to permutation of the table content resulting from such operations**. Besides, Hytrel also does not allow arbitrary invalid cell permutations (arbitrarily swap of two cells in a table), which will destroy the table structure and cause excessive invariance problems, as we have emphasized in Appendix B2.
>
>
>
> **Q3**: Or, in the case where row/column permutation is the main target, it would be helpful to show how it negative affects downstream tasks, motivating the development of an permutation invariant encoding representation.
>
> **A3**: As for the relationship between row/column permutations and downstream tasks. We experiment with different permutations on the TTD benchmark with the TaBERT baseline (which does not model the permutation invariance property) and our approach, as in the following table.
>
> ||Table Type Detection (TTD, Accuracy, %) ||| |
> |-|-|-|-|-|
> || No Permutation | Row Permutation | Column Permutation | Row & Column Permutations |
> |TaBERT (K=1) | 93.11 | 93.13 | 92.67 | 92.64 |
> |TaBERT (K=3) | 95.15 | 95.03 | 94.61 | 94.31|
> | HyTrel - ELECTRA | 95.81 | 95.81 | 95.81 | 95.81
> | HyTrel - Contrast  | 94.52 |  94.52 | 94.52 |  94.52 |
>
> We observe that the performance of our HyTrel models will not be affected by the permutation because the representations are the same with the permutations. However, we can see that for the TaBERT model, different permutation types will degrade the performance a little bit, and the combination of the permutations hurt the performance most.
>
> Please note that:
>
> a. TaBERT K=1 with only one row, the row permutation is equivalent to resample a different row, so the performance is very close to the one without permutations.
>
> b.  Each model is evaluated with the same parameter settings for different permutation types.

---

> > ### Comment · Reviewer_kUWt · 2023-08-21
> >
> > Thanks to the authors for providing additional explanations and running additional experiments. I have raised my score.
> >
> > As a side note, it seems like the gain from the permutation invariance property is rather limited, not sure if it is due to the saturation of the task or the simplicity of the database/tables. Looking forward to the extension of this work being adapted to multi-table situations.

---

> > > ### Author Response · Authors · 2023-08-21
> > >
> > > Thank you for revisiting our work and considering our response; we appreciate your willingness to raise the score. We will dive into more about the permutation invariance benefits and multi-table situations.

---

### Official Review · Reviewer_QZu3 · 2023-07-07

**Soundness:** 3 good
**Presentation:** 3 good
**Contribution:** 2 fair
**Rating:** 6
**Confidence:** 1

**Summary:**

In the paper, the authors aim to capture the structural properties of tabular data using hyper-graphs with four different types of hyperedges based on the co-occurrences in the table. The experimental results on four downstream tasks show the advantages of the proposed method on other competitive baselines.

**Strengths:**

The manuscript is well-structured and easy to follow. The overall idea of modeling high-order cell/rows/columns interactions using hyper-graph is novel.
1.	The proposed hyper-graph encoder captures tabular structures via attention learning.
2.	The experimental results, which extensive ablation studies, convincingly demonstrate the advantages of the proposed methods.
3.	The visualization in Sec. 4.2 shows that hierarchical representation can be learned by hyper-graph.


**Weaknesses:**

Minors
1.	Since hyper-graph excels majorly in handling hyper-edges of varied orders. It would be sub-optimal for hyper-graph to handle the hype-edges of fixed order in the tabular data?
2.	It’s expected that some baseline hyper-graph models are included in the compassion results such as [1,2].
3.	It’s suggested to elaborate on the hierarchical structures of tabular data in the Introduction otherwise a little bit confusing.    Further, it's still unclear the relationship between hierarchical structures and hypergraphs.

[1] Hypergraph Neural Networks, 2018
[2] Hypergraph convolution and hypergraph attention, 2020


**Questions:**

NON

**Limitations:**

NON

---

> ### Author Rebuttal · Authors · 2023-08-10
>
> **Q1**: Since hyper-graph excels majorly in handling hyper-edges of varied orders. It would be sub-optimal for hyper-graph to handle the hype-edges of fixed order in the tabular data?
>
> **A1**: We do not fix the orders of hyperedges. The **orders of hyperedges depend on the size of the tables**- and are different for different tables, as different tables have different numbers of columns and rows. For the hypergraph built from a given table with $m$ rows and $n$ columns, the order of the row hyperedge is $n$, the order of the column hyperedge is $m$, and the order of the table hyperedge is $mn$. So the same type of hyperedges have the same order, but the hyperedge order is not fixed for the whole table.
>
> As for the sub-optimality of a hypergraph that has fixed order hyperedges - unfortunately we are unaware of any such work (theoretical or practical which raise this concern). We would appreciate it if you could kindly refer to us to works which state this - and we will appropriately incorporate this into the manuscript.
>
> **Q2**: It’s expected that some baseline hyper-graph models are included in the compassion results such as [1,2].
>
> [1] Hypergraph Neural Networks, 2018
>
> [2] Hypergraph convolution and hypergraph attention, 2020
>
> **A2**: It is important to note that that primary goal of our work was to model and showcase the benefits of modeling table as hypergraphs rather than designing new hypergraph neural networks, We choose AllSet -- the set-transformer based Hypergraph Neural Network because **[3] demonstrated that it is provably more expressive than HNN [1] or the HCHA [2]**. Specifically, [3] points out that the set transformer they leverage can capture high-order relations amongst nodes within the same hyperedge. On the contrary, HNN and HCHA use a clique-expansion approach thereby losing any such higher order interaction information. We leave the experiments with HNN or HCHA in our frameworks as future work.
>
> [3] Chien, Eli, et al. "You are allset: A multiset function framework for hypergraph neural networks." arXiv preprint arXiv:2106.13264 (2021).
>
> **Q3**: It’s suggested to elaborate on the hierarchical structures of tabular data in the Introduction otherwise a little bit confusing. Further, it's still unclear the relationship between hierarchical structures and hypergraphs.
>
> **A3**: We apologize for any confusion in the introduction. The hierarchical structure of a table is that we can learn the representations or semantics of a whole table from the sub-level rows and columns semantics, and the rows and columns semantics can further be aggregated from its constituent cell values. So **we can learn coarse grained representations of table elements like row/s, column/s, tables using finer grained cell representations through aggregation functions**. Previous BERT-based TaLMs abstract this by serializing the content of these cells into tokens in some order and using the BERT encoder as the aggregation function. But these methods ignore the innate table structure like the permutation invariance etc - which we capture by modeling the table as a hypergraph, and use the HyperTrans model as the aggregation functions to aggregate the fine-grained cell information into the coarse-grained row/column and the whole table.

---

> > ### Comment · Reviewer_QZu3 · 2023-08-17
> >
> > Greatly appreciate the clarifications from the reviewers.  I would like to keep the score.

---

> > > ### Author Response · Authors · 2023-08-19
> > >
> > > Thank you for your consideration and the time you've taken to review our work. We appreciate your feedback.

---

### Author Rebuttal · Authors · 2023-08-10

We sincerely thank all the reviewers for their thoughtful feedback. We address the reviewers' comments below individually and will incorporate all the feedback in our paper.

---

### Decision · Program_Chairs · 2023-09-21

**Decision:**

Accept (spotlight)

**Comment:**

The reviewers felt that the manuscript was well-written, addressing an important question, with good evidence of the benefit of the method. The back and forth between authors and reviewers made the submission stronger.